# Deletion of Macrophage-Specific Glycogen Synthase Kinase (GSK)-3α Promotes Atherosclerotic Regression in *Ldlr*^−/−^ Mice

**DOI:** 10.3390/ijms23169293

**Published:** 2022-08-18

**Authors:** Sarvatit Patel, Lauren Mastrogiacomo, Madison Fulmer, Yuanyuan Shi, Geoff H. Werstuck

**Affiliations:** 1Thrombosis and Atherosclerosis Research Institute, 237 Barton Street E, Hamilton, ON L9L 2X2, Canada; 2Department of Chemistry and Chemical Biology, McMaster University, 1280 Main Street W, Hamilton, ON L8S 4L8, Canada; 3Department of Medicine, McMaster University, 1280 Main Street W, Hamilton, ON L8S 4L8, Canada

**Keywords:** atherosclerosis, glycogen synthase kinase (GSK)-3α/β, inflammation, macrophages, mice model, migration, regression, reverse cholesterol transport

## Abstract

Recent evidence from our laboratory suggests that impeding ER stress–GSK3α/β signaling attenuates the progression and development of atherosclerosis in mouse model systems. The objective of this study was to determine if the tissue-specific genetic ablation of GSK3α/β could promote the regression of established atherosclerotic plaques. Five-week-old low-density lipoprotein receptor knockout (*Ldlr*^−/−^) mice were fed a high-fat diet for 16 weeks to promote atherosclerotic lesion formation. Mice were then injected with tamoxifen to induce macrophage-specific GSK3α/β deletion, and switched to standard diet for 12 weeks. All mice were sacrificed at 33 weeks of age and atherosclerosis was quantified and characterized. Female mice with induced macrophage-specific GSK3α deficiency, but not GSK3β deficiency, had reduced plaque volume (~25%) and necrosis (~40%) in the aortic sinus, compared to baseline mice. Atherosclerosis was also significantly reduced (~60%) in the descending aorta. Macrophage-specific GSK3α-deficient mice showed indications of increased plaque stability and reduced inflammation in plaques, as well as increased CCR7 and ABCA1 expression in lesional macrophages, consistent with regressive plaques. These results suggest that GSK3α ablation promotes atherosclerotic plaque regression and identify GSK3α as a potential target for the development of new therapies to treat existing atherosclerotic lesions in patients with cardiovascular disease.

## 1. Introduction

Cardiovascular disease is one of the major causes of death in the world today [1]. Atherosclerosis is a major underlying cause of both cardio- and cerebrovascular disease. Atherosclerosis involves the build-up of fatty plaques in the arteries making them hard and narrow, thereby restricting blood flow and potentially damaging the heart or peripheral tissues [2]. Atherosclerotic plaques initiate in areas with bifurcations, branches, or inner curvatures where the blood flow is turbulent and non-laminar [3]. At these sites, circulating monocytes and T lymphocytes migrate into the sub-endothelial intimal layer [4,5]. In the sub-endothelium, monocytes differentiate into macrophages. Macrophages endocytose modified low-density lipoprotein (LDL) particles and become foam cells. These foam cells make up the earliest discernable lesion, called the fatty streak. Macrophage foam cells and lymphocytes secrete cytokines and growth factors, including interferon (IFN)-γ, interleukin (IL)-1β, and tumor necrosis factor (TNF)-α, which amplify the inflammatory response [6,7]. Due to a combination of apoptosis and defective efferocytosis in advanced lesions, an acellular region appears within the lesion, known as the necrotic core. Necrosis is a key feature of unstable plaques that are prone to rupture. Plaque rupture can result in acute cardiovascular complications, such as myocardial infarction or stroke, and potentially death.

Atherosclerotic plaques can regress or shrink in size. Intervention studies have shown changes in plaque characteristics including reduced total atheroma volume, in response to various treatments [8]. It is now clear that this is not just the simple mechanistic reversal of atherosclerotic progression. Regressive and/or stable plaques are characterized by reduced macrophage content, increased vascular smooth muscle cell (VSMC) content, thicker fibrotic cap material, reduced apoptosis, and smaller necrotic cores. There are five potentially important cellular mechanisms that can contribute to plaque regression [9,10,11,12,13,14,15]. These include: (i) reverse cholesterol transport (RCT), in which cholesterol and lipids are transported out of lesional macrophage/foam cells; (ii) macrophage egression out of the plaque [10,11]; (iii) reduced monocyte recruitment into the plaque; (iv) increased ratio of anti-inflammatory to pro-inflammatory macrophages (M2/M1) [12,13,14,15]; and (v) reduced macrophage proliferation within the plaque [9]. Likely, each of these processes play a role in the regression of atherosclerotic lesions. However, the underlying molecular mechanisms that regulate macrophage function during the atherosclerotic regression are not completely understood.

Our lab and others have previously shown that vascular endoplasmic reticulum (ER) stress activates glycogen synthase kinase (GSK3) to promote atherosclerotic development [16,17,18]. GSK3 is a serine/threonine kinase that plays an important role in many cellular pathways that regulate metabolism and viability. GSK3 has been linked to several disorders and diseases, such as cancer [19], bipolar mood disorder [20], diabetes [21], and Alzheimer’s disease [22]. There are two forms of GSK3 in mammals: GSK3α (51 kDa) and GSK3β (47 kDa), as well as the splice variant of GSK3β, GSK3β2 [23]. Isoforms GSK3α and GSK3β are 98% homologous in the kinase domain and are expressed ubiquitously [24]. The inhibition of GSK3α/β reduces the uptake of free and modified cholesterol, as well as the expression of genes involved in regulating lipid and cholesterol metabolism [18]. Targeting the ER stress–GSK3α/β pathway, either genetically or pharmacologically can attenuate the progression and development of atherosclerosis in a mouse model system [17,25,26].

It is now clear that, while GSK3α and GSK3β do share many substrates, they also play distinct roles within the cell [27,28,29]. Results from our lab suggest that myeloid deletion of GSK3α, but not GSK3β, attenuates the progression of atherosclerosis [30]. Furthermore, recent data suggest that myeloid GSK3α and GSK3β have distinct effects in the regulation of macrophage phenotype, inflammatory response, lipid accumulation, and migration [31]. However, the potential role of macrophage-specific GSK3α and GSK3β in atherosclerotic regression is not known.

We have developed inducible macrophage-specific GSK3α and/or GSK3β knockout low-density lipoprotein receptor (*Ldlr*)^−/−^ mice to examine the roles of GSK3α/β in atherosclerosis. In these studies, we utilize this model to investigate the effects of GSK3α/β ablation on existing atherosclerotic plaques.

## 2. Results

### 2.1. Characterization of Tamoxifen-Induced Macrophage-Specific GSK3α and/or GSK3β Deficiency in Ldlr^−/−^ Mice

Novel mouse strains that facilitate inducible, tissue-specific GSK3α and/or GSK3β deletion were used to investigate the potential to promote atherosclerotic regression (Appendix A). The tamoxifen-induced macrophage-specific ablation of GSK3α and/or GSK3β expression was confirmed by immunoblot and RT-PCR (Appendix A). Compared to untreated control mice, macrophages derived from tamoxifen-treated LiαKO and LiβKO mice show significantly reduced protein (Appendix A) and mRNA (Appendix A) expression of GSK3α and GSK3β, respectively. Tamoxifen treatment did not significantly affect GSK3α or GSK3β protein expression in other tissues, including muscle and liver of LiαKO, LiβKO, or LiαβKO mice (Appendix A). This result indicates that the induced GSK3α/β ablation is specific to macrophages.

### 2.2. Macrophage-Specific GSK3α and/or GSK3β Deficient Ldlr^−/−^ Mice Are Viable and Develop Normally

Baseline mice had significantly higher total plasma cholesterol and triglyceride levels than all other experimental groups. No significant differences were observed between experimental LiαKO, LiβKO, and LiαβKO mice treated with or without tamoxifen (Appendix A). There was no significant difference in fasting blood glucose, body weight or liver weight between experimental groups comprised of female (Appendix A) or male mice (Appendix A). Tamoxifen-treated female mice had significantly lower adipose tissue weight compared to controls (Appendix A). No differences were observed in adipose tissue weight of male mice.

### 2.3. Macrophage-Specific GSK3α-, but Not GSK3β-, Deficiency Promotes Atherosclerosis Plaque Regression in Ldlr^−/−^ Mice

To determine if GSK3α and/or GSK3β deletion can induce plaque regression, we established atherosclerotic plaques in LiαKO, LiβKO, and LiαβKO mice by HFD feeding for 16 weeks. At this time, the baseline group was sacrificed and a subset of mice from each remaining group were treated with tamoxifen to ablate expression of GSK3α and/or GSK3β. All mice were then fed a standard chow diet for 12 wks (Appendix A).

To investigate the effects of GSK3α and/or GSK3β deletion on atherosclerosis, the aortic sinus was removed and processed, as previously described [32]. Cross-sections of the aortic root were stained with Masson’s Trichrome stain (Figure 1A) and the atherosclerotic lesion area and volume were quantified (Figure 1B–D). The results show that female LiαKO and LiαβKO mice treated with tamoxifen had reduced lesion area and lesion volume compared to the same strains that were not treated with tamoxifen (Figure 1B,D). In addition, LiαKO and LiαβKO mice treated with tamoxifen showed reduced plaque volume compared to baseline (Figure 1B,D). The lesion area and volume in LiβKO mice was not different than the respective untreated controls or baseline (Figure 1C). Male LiαKO and LiαβKO mice treated with tamoxifen showed reduced atherosclerotic lesion area and volume compared to the respective untreated control group, but not baseline (Appendix A). Male LiβKO mice treated with tamoxifen showed no difference in lesion area and volume compared to the respective untreated controls or baseline (Appendix A). The tamoxifen treatment itself had no effect on lesion area or volume in negative control mice (Csf1riCre^−/−^) (Appendix A). Together, these data suggest that macrophage-specific removal of GSK3α can promote atherosclerosis plaque regression in female *Ldlr*^−/−^ mice.

Lesional necrotic core area was determined by trichrome staining of aortic cross sections from each of the experimental groups. Compared to baseline and the respective untreated controls, female LiαKO and LiαβKO mice treated with tamoxifen showed reduced necrotic core as a percentage to total plaque area at the aortic sinus (Figure 1B,D). LiβKO mice treated with tamoxifen showed no changes in relative necrotic core compared to the respective untreated controls or baseline (Figure 1C). Similar effects were observed in male mice (Appendix A). As reduced necrotic core size is one of the indications of plaque regression, this result is consistent with GSK3α ablation promoting atherosclerotic plaque regression.

En face aortas from female baseline, LiαKO, LiβKO, and LiαβKO mice were stained with Sudan IV to quantify atherosclerotic plaques in total aorta (Figure 2A). LiαKO and LiαβKO mice treated with tamoxifen had significantly less atherosclerotic lesion area across the entire aorta (aortic arch and thoracic aorta) compared to baseline and the respective untreated control groups (Figure 2B–D). LiβKO mice treated with tamoxifen showed no difference in en face lesion area compared to the respective untreated controls or baseline (Figure 2B–D).

Collectively, these data suggest that macrophage-specific GSK3α, but not GSK3β, ablation can impede the development of existing atherosclerosis and promote atherosclerotic regression in female Ldlr^−/−^ mice. Further characterization of atherosclerotic lesions was carried out in the female mouse model.

### 2.4. Macrophage-Specific GSK3α-Deficiency Promotes a Phenotype Associated with Increased Atherosclerotic Plaque Stability in Female Ldlr^−/−^ Mice

To further explore the effects of GSK3α/β deficiency on the stability of atherosclerotic plaques, properties including smooth muscle cell content covering the fibrous cap and macrophage contribution to plaque composition were examined. Lesional vascular smooth muscle cell and macrophage/foam cell content were quantified in each of the experimental groups using immunofluorescent staining with antibodies against smooth muscle α actin and CD107b (Mac-3), respectively (Figure 3A,B). Tamoxifen-treated LiαKO mice had significantly increased lesional smooth muscle content compared to untreated controls and baseline (Figure 3C). Baseline mice showed elevated lesional macrophage content relative to all experimental groups, indicative of an effect that can be attributed to differences in diet composition (Figure 3D). Tamoxifen-treated LiαKO mice had significantly lower lesional macrophage content compared to untreated controls (Figure 3D). LiβKO mice treated with tamoxifen showed no changes in vascular smooth muscle cell or macrophage/foam cell content compared to the respective untreated controls or baseline (Figure 3C,D). As previously described [33], plaque stability was assessed by quantifying the ratio of SMC area to macrophage area. Results suggest that plaques of tamoxifen-treated LiαKO mice are significantly more stable than the respective controls or baseline (Figure 3E). This evidence indicates that macrophage-specific GSK3α-deletion plays a protective role in advanced atherosclerotic lesions by enhancing several aspects of atherosclerotic plaque stability at the aortic root.

### 2.5. Macrophage-Specific GSK3α Deficiency Reduces the Pro-Inflammatory Response

Pro-inflammatory responses within the plaques were determined by performing immunofluorescence staining using antibodies against NF-κB p65, NLRP3, and IL-1β (Figure 4A–C). The tamoxifen-treated LiαKO group had significantly lowered NF-κB p65 expression compared to the untreated LiαKO group (Figure 4D). The tamoxifen-treated LiαKO group had significantly lower NLRP3 and IL-1β expression compared to the untreated LiαKO group and baseline (Figure 4E,F). LiβKO mice treated with tamoxifen showed no changes in NF-κB p65, NLRP3, or IL-1β expression compared to the corresponding untreated controls or baseline (Figure 4D,F). These results suggest that the activation of proinflammatory response pathways is dampened in macrophage-specific GSK3α-deficient mice.

### 2.6. Elevated ABCA1 Expression in GSK3α-Deficient Macrophages

Atherosclerotic regression is associated with increased RCT from macrophage/foam cells residing in the plaque^9^. Markers of RCT in lesional macrophages were quantified by performing immunofluorescence co-staining of aortic cross sections from each of the experimental groups with antibodies against ABCA1, a marker of RCT, and CD68 (Figure 5A). All experimental groups, with or without tamoxifen treatment, showed significantly decreased expression of cholesterol efflux regulator ABCA1 compared to baseline (Figure 5B). Tamoxifen-treated LiαKO mice had significantly lower macrophage (CD68^+^) content compared to untreated controls and baseline (Figure 5C). This finding is consistent with our previous results showing that the tamoxifen-treated LiαKO group had significantly lower lesional macrophage content compared to the untreated control group and baseline. Tamoxifen-treated LiαKO mice have significantly more ABCA1^+^ macrophages (ABCA1^+^CD68^+^), compared to untreated controls and baseline (Figure 5D). These findings are consistent with atherosclerotic regression in macrophage-specific GSK3α-deficient mice.

### 2.7. GSK3α-Deficient Mice Have Elevated CCR7 Expression

Another mechanism of atherosclerotic regression involves CCR7-dependent macrophage egress from the plaque [34,35]. The propensity of lesional macrophages to egress out of plaque was determined by performing immunofluorescence co-staining in each of the experimental groups using antibodies against CCR7, a marker of migration, and CD68 (Figure 6A). Baseline mice showed significantly elevated lesional CCR7 expression relative to all experimental groups (Figure 6B). The tamoxifen treated LiαKO group had significantly lower macrophage (CD68^+^) content compared to the untreated LiαKO group and baseline (Figure 6C). This finding is consistent with our observation that tamoxifen-treated LiαKO mice had significantly decreased lesional macrophage content compared to untreated controls and baseline. When normalized to macrophage content, tamoxifen-treated LiαKO mice had significantly increased CCR7^+^ macrophage (CCR7^+^CD68^+^) content compared to untreated controls and baseline (Figure 6D). This result is consistent with an association between macrophage-specific GSK3α-deficiency and increased atherosclerotic regression.

## 3. Discussion

In this study, we utilized a novel, macrophage-specific, inducible GSK3α/β knockout mouse model to investigate the role of GSK3α/β in atherosclerotic regression. We demonstrated that macrophage-specific deletion of GSK3α, but not GSK3β, promotes the regression of established atherosclerotic plaques in *Ldlr*^−/−^ mice. Analysis of aortic root shows that this regression is associated with increased plaque stability and decreased pro-inflammatory response. Furthermore, examination of the mechanisms that contribute to atherosclerotic plaque regression show that macrophage-specific GSK3α deficiency is associated with increased expression of the RCT marker (ABCA1) and migration marker (CCR7) in lesional macrophages. Together, these data show that macrophage-specific deletion of GSK3α promotes an anti-atherogenic macrophage phenotype resulting in significant regression of the atherosclerotic plaque.

Previous studies from our lab and others have established a role for ER stress–GSK3α/β signaling in the progression and development of atherosclerotic plaques in mouse models. Specifically, we showed that genetic ablation of myeloid GSK3α, or pharmacological inhibition of GSK3α/β, can significantly impede growth of atherosclerotic lesions in both *ApoE*^−/−^ and *Ldlr*^−/−^ mouse strains [17,25,26,30,36]. Clinically useful anti-atherogenic interventions will require that the therapies be effective at shrinking, or at least stabilizing, established atherosclerotic plaques. Therefore, to test the effect of GSK3α/β ablation on established lesions, we created novel *Ldlr*^−/−^ mouse strains that allow for inducible, tissue-specific GSK3α and/or GSK3β deletion using Cre-lox technology (Appendix A). Upon tamoxifen injection, these mice show macrophage-specific deletion of GSK3α and/or GSK3β in respective experimental groups. GSK3α/β expression was not significantly affected in other tissues (skeletal muscle, liver).

In order to establish advanced atherosclerotic plaques in this model, mice were fed an HFD for 16 weeks. The baseline subgroup was sacrificed at this time to permit measurement and characterization of the established lesions. All other experimental groups were switched to chow diet with or without tamoxifen treatment to delete macrophage GSK3α/β. After an additional 12 weeks on standard chow diet, all remaining mice were sacrificed and analyzed.

Baseline mice had significantly elevated total plasma cholesterol and triglyceride concentration compared to all other experimental groups. This difference is due to the fact that baseline mice were only fed HFD, whereas all other groups were switched to standard diet. Tamoxifen-treated female mice had significantly lower adipose tissue weight compared to the untreated mice. There was no difference in adipose tissue weight in male mice. This result supports previously published data, showing that tamoxifen can reduce subcutaneous and visceral white adipose tissue accumulation in mice, and suggests that female mice are more susceptible to this effect [37,38].

GSK3α-deficient female mice had reduced lesion area and lesion volume in the aortic sinus and whole aorta compared to baseline and untreated controls. The results also show that tamoxifen treatment had no effect on lesion area or volume in negative control mice (*Csf1riCre*^−/−^), indicating that tamoxifen itself does not have a direct effect on the atherosclerotic plaque.

In male mice, the induced macrophage-specific GSK3α deficiency was associated with reduced atherosclerotic lesion area and volume compared to the respective untreated control group, but not baseline. While these results are consistent with previous observations, indicating that the macrophage-specific GSK3α deficiency can impede the development of existing atherosclerosis in *Ldlr*^−/−^ mice, it is not possible to make conclusions regarding plaque regression in male mice because the plaques continued to grow after the switch to standard chow diet (compare baseline to control). This sex difference is likely a result of the male mice having significantly smaller plaques at baseline (after 16wks of HFD), relative to female mice (Appendix A). It is recognized that female *Ldlr*^−/−^ mice develop significantly larger plaques, compared to males, after 14, 18, and 22 weeks on the HFD [39], and it is likely that the stage of plaque development is a factor on the ability to arrest plaque growth by dietary switch. Because of this observed sex difference, the majority of the subsequent analysis was performed on the regressive plaques of female mice.

The results of characterization of the atherosclerotic plaques in the female GSK3α-deficient mice are consistent with regression/stabilization. Previous studies have shown that atherosclerotic regression is associated with an upregulation of ABCA1 expression and enhanced RCT in lesional macrophages of *Ldlr*^−/−^ mice treated with an antisense to micro-RNA-33 [40]. Other studies suggest that mice with macrophage-specific low-density lipoprotein-receptor-related protein 1 (LRP1) deficiency or statin treatment showed increased CCR7 expression in lesional macrophages to promote the atherosclerosis regression [34,41]. In this study, we observed increased expression of ABCA1 and CCR7 in lesional macrophages of the GSK3α-deficient mice, consistent with increased reverse cholesterol transport and mobilization of macrophages, which would be expected in a regressive plaque. These mice also show reduced macrophage (Mac3^+^ or CD68^+^) content and significantly smaller necrotic cores, compared to baseline and control mice, which may be indicative of macrophage egress out of the plaques. The retained macrophages express significantly lower levels of pro-inflammatory factors including NF-κB, NLRP3, and IL-1β, which suggests a switch to an M2-like phenotype.

Together, these studies suggest that GSK3α plays an active role in promoting an inflammatory phenotype in lesional macrophages that promotes atherosclerotic progression and impairs regression. Cardiovascular risk factors, including diabetes mellitus and hypercholesterolemia, have been shown to hypoactivate Akt signaling, which resulted in increased inflammation, cellular proliferation, apoptosis, and vasa vasorum neovascularization, resulting in advanced atherosclerosis [42]. GSK3α/β is a direct substrate of Akt and is inactivated by this kinase. It is known that GSK3α is an upstream regulator of several substrates, including NF-κB, STAT, and β-catenin, which influences polarization and atherogenic functions of macrophages [30,31,43]. Recent evidence suggests that GSK3α, but not GSK3β, modulates the JAK-STAT pathway by indirectly affecting STAT phosphorylation [30]. Previously we have shown that GSK3α specifically regulates STAT3/6 phosphorylation/activation and NF-κB expression in macrophages [30,31]. This suggests that GSK3α, but not GSK3β, actively plays a role in M1 macrophage polarization and pro-inflammatory response. The upstream mechanisms by which GSK3α and GSK3β may be differentially regulated are not fully understood. The further elucidation of this pathway will shed light on the mechanisms relevant to pro- and anti-atherosclerotic processes and potential targets for therapeutic intervention.

Previous studies from our lab have shown that supplementation of *Ldlr*^−/−^ mice with valproate, a small molecule with pan GSK3α/β inhibitory properties, showed indication of more stable atherosclerotic plaques [25,26]. Isoform-specific inhibitors of GSK3α/β have recently been identified [44]. The use of these specific inhibitors is a critical next step in determining the potential of GSK3α as a drug target in the treatment of atherosclerotic CVD. Further investigation will be needed to define the basic molecular/cellular mechanisms and to delineate downstream signaling pathways.

In summary, these studies have revealed a novel mechanism by which macrophage-specific ER stress–GSK3α signaling promotes atherosclerotic development and impairs resolution/regression of plaques. Ablation of GSK3α alters the macrophage phenotype and promotes atherosclerotic regression. Together, these results suggest that GSK3α may be a viable target for the development of anti-atherogenic therapies.

## 4. Materials and Methods

### 4.1. Mouse Models

We created an inducible knock out mouse model in which we can trigger the deletion of GSK3α and/or GSK3β at a specific time using Cre-lox technology (Appendix A). *Ldlr*^−/−^ mice carrying a loxP-flanked GSK3α gene (*Ldlr^−/−^GSK3α^fl/fl^*) were crossed with mice expressing a single copy of the Csf1riCre recombinase gene controlled by the colony stimulating factor 1 receptor (Csf1r) promoter (*Ldlr^−/−^Csf1riCre^+/−^GSK3α^fl/fl^*). Using this breeding method, we were able to generate the *Ldlr*^−/−^ macrophage-specific GSK3α inducible knockout mice (*Ldlr^−/−^Csf1riCre^+/−^GSK3α^fl/fl^* or LiαKO) (Appendix A). The *Ldlr*^−/−^ macrophage-specific GSK3β deficient mice were similarly crossed to obtain *Ldlr*^−/−^ macrophage-specific GSK3β inducible knockout mice (*Ldlr^−/−^Csf1riCre^+/−^GSK3β^fl/fl^* or LiβKO) (Appendix A). The above mice were crossed to generate *Ldlr*^−/−^ macrophage-specific GSK3α/β inducible knockout mice (*Ldlr^−/−^Csf1riCre^+/−^GSK3α^fl/fl^GSK3β^fl/fl^* or LiαβKO) (Appendix A). All the mouse strains described above exist in a C57Bl/6 genetic background. All animal experiments were preapproved by the McMaster University Animal Research Ethics Board. All experiments conform with the guidelines and regulation of the Canadian Council on Animal Care.

### 4.2. Atherosclerotic Regression Model

Five-week-old male and female mice (baseline, LiαKO, LiβKO, and LiαβKO) were fed a high-fat diet (HFD) containing 21% fat and 0.2% cholesterol, with 42% calories from fat for 16 weeks to establish atherosclerotic plaques (Appendix A). Mice were injected intraperitoneally with tamoxifen (5 mg/100 µL/day) for 3 days. Control mice received corn oil. Baseline mice were harvested after 16 weeks of HFD feeding. All other mice were switched to a chow diet containing 18% protein and 5% fat, with 18% calories from fat to arrest atherosclerotic development. These mice were harvested after 12 weeks on the chow diet (33 weeks of age). All mice were granted free access to water.

Mice were fasted for 6 h prior to sacrifice. Body weight was measured, and 3% isoflurane was used to anesthetize mice. Blood was collected via cardiac puncture and livers and perigonadal adipose tissue were harvested and weighed. The vasculature was flushed with phosphate buffered saline (PBS) and perfusion fixed with 10% neutral buffer formalin. Hearts and aortas along with other tissues were collected and fixed in formalin.

### 4.3. Determination of Plasma Lipids

Total plasma cholesterol and triglyceride levels were determined by using the Infinity Reagent Cholesterol kit (TR13421) or Infinity Reagent Triglyceride kit (TR22421), respectively. Assays were performed according to manufacturer’s instructions.

### 4.4. Immunoblot

Cell or tissue extracts were fractionated by SDS-PAGE and transferred to a polyvinylidene difluoride (PVDF) membrane (Bio-Rad, Mississauga, ON, Canada). The membrane was incubated with 5% non-fat milk in TBST (10 mM Tris, pH 8.0, 150 mM NaCl, 0.5% Tween 20) for 45 min and washed with TBST for 5 min. The membrane was then incubated with a primary antibody against GSK3α/β (1:1000) [Cell signaling, Whitby, ON, Canada] or β-Actin (1:3000) [Sigma, Oakville, ON, Canada], at 4 °C overnight. The next day, membranes were washed three times for 5 min and incubated with a horseradish peroxidase-conjugated anti-rabbit antibody (1:200) [Agilent Dako, Mississauga, ON, Canada] or anti-mouse antibody (1:200) [Agilent Dako, Mississauga, ON, Canada] for 1 h. Blots were washed with TBST three times for 5 min each and developed with the ECL system (Millipore, Oakville, ON, Canada). Images were captured and quantified using a Molecular Imager ChemiDoc XRS+ (Bio-Rad, Mississauga, ON, Canada).

### 4.5. Characterization of Aortic Lesions

Hearts and aortas were embedded in paraffin and 5 μm sections were collected onto slides, starting from the aoric sinus and moving up the ascending aorta [32]. Sections were stained with Masson’s Trichrome (Sigma, Oakville, ON, Canada) for atherosclerotic lesion quantification at aortic sinus. Images of the stained sections were captured using Olympus BX41 microscope connected to a DP71 Olympus camera. Lesion area was quantified using Image J 1.48v software (Wayne Rasband, Online).

For immunofluorescent staining, sections were deparaffinized and antigen retrieval performed using antigen unmasking solution (Vector laboratories, Brockville, ON Canada). Sections were blocked in 10% goat serum and then immunostained overnight with primary antibodies against the macrophage marker Cluster of Differentiation (CD) 107b (Mac3) at a dilution of 1:50 (BD Transductions, United States), CD68 diluted 1:100 (Thermofisher Scientific Mississauga, ON, Canada), VSMC marker α-actin diluted 1:100 (Santa Cruz, Dallas, TX, United States), pro-inflammatory marker Nuclear factor kappa B (NF-κB p65) diluted 1:50 (Santacruz, Dallas, TX, United States), IL-1β diluted 1:100 (Invitrogen, Mississauga, ON, Canada), inflammasome marker NLR Family Pyrin Domain Containing 3 (NLRP3) diluted 1:100 (Invitrogen, Mississauga, ON, Canada), macrophage migration marker C-C Motif Chemokine Receptor 7 (CCR7) diluted 1:100 (abcam, Waltham, MA, United States), or RCT marker ATP Binding Cassette Subfamily A Member 1 (ABCA1) diluted 1:100 (NOVUS, Toronto, ON, Canada). Sections were then incubated with secondary antibodies Alexa Fluor 488 goat anti-mouse IgG diluted 1:250 (Thermofisher Scientific, Mississauga, ON, Canada), Alexa Fluor 488 goat anti-rabbit IgG diluted 1:250 (Thermofisher Scientific, Mississauga, ON, Canada), Alexa Fluor 488 goat anti-rat IgG diluted 1:250 (Thermofisher Scientific, Mississauga, ON, Canada), or Alexa Fluor 594 goat anti-rat IgG diluted 1:250 (Thermofisher Scientific, Mississauga, ON, Canada) for 2 h, and then stained with the DAPI diluted 1:5000 (Invitrogen, Mississauga, ON, Canada). Slides were mounted with Fluoromount Aqueous Mounting Medium (Sigma, Oakville, ON, Canada) and stored at 4 °C in the dark. Separate slides were stained with pre-immune IgG instead of primary antibodies to control for non-specific staining (Appendix A). Images of the stained sections were collected using the Leica STELLARIS 5 confocal microscope. Image J 1.52q software was used to quantify immunofluorescent staining.

### 4.6. Characterization of En Face Aortic Lesions

Fixed whole aortas were isolated and cleaned of surrounding muscle and adventitial fat. Whole aortas were longitudinally dissected and stained for lipid content with Sudan IV (Sigma, Oakville, ON, Canada). En face images of the whole aorta were captured using a digital camera (Canon EOS Rebel T7). The percentage of the atherosclerotic area was assessed by quantifying stained areas divided by the entire surface area using Image J 1.52q software.

### 4.7. Statistical Analysis

All statistical analyses were performed in GraphPad Prism software (version 9.3.1). All data was analyzed by a one-way ANOVA, followed by the Tukey’s multiple comparison test between all groups. All error bars on graphs represent the standard error of the mean (SEM). For all experiments, a *p* value lower than 0.05 was considered statistically significant. * *p* < 0.05, ** *p* < 0.01, *** *p* < 0.001, and **** *p* < 0.0001.

## Figures and Tables

**Figure 1 ijms-23-09293-f001:**
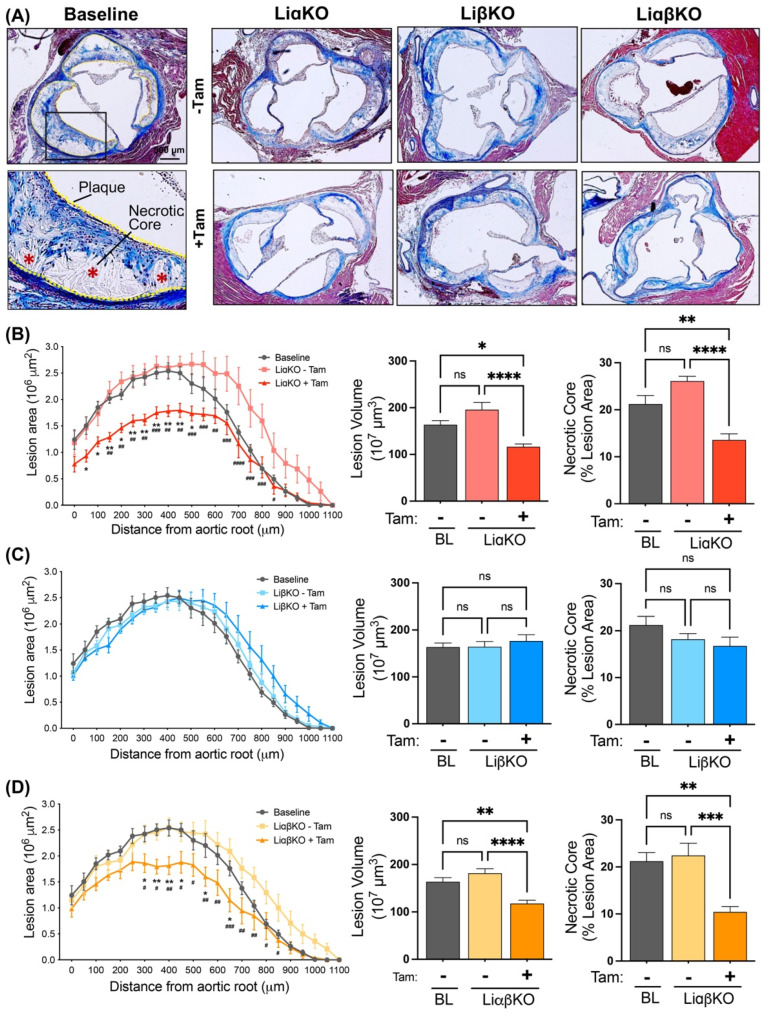
Macrophage-specific GSK3α-, but not GSK3β-, deficiency promotes atherosclerosis plaque regression in female Ldlr^−/−^ mice. (**A**) Representative Masson’s trichrome stained aortic root sections from high-fat diet fed baseline, LiαKO, LiβKO, and LiαβKO mice with and without tamoxifen treatment. Quantification of atherosclerotic lesion area, lesion volume, and % necrotic area at the aortic sinus and ascending aorta in (**B**) LiαKO, (**C**) LiβKO, and (**D**) LiαβKO mice. n = 9–13; mean ± SEM; * represents the comparison to baseline; * *p* < 0.05, ** *p* < 0.01, *** *p* < 0.001, **** *p* < 0.0001 and ^#^ represents the comparison between −Tam and +Tam; ^#^ *p* < 0.05, ^##^ *p* < 0.01, ^###^ *p* < 0.001, ^####^ *p* < 0.0001 (one-way ANOVA). Tam, tamoxifen; BL, baseline. ns, not significant.

**Figure 2 ijms-23-09293-f002:**
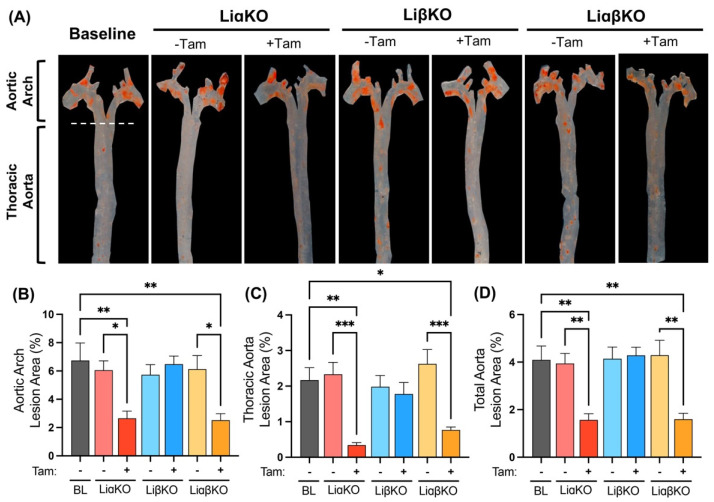
En face determination of lesional area in the whole aorta of tamoxifen induced macrophage-specific GSK3α and/or GSK3β deficient female Ldlr^−/−^ mice. (**A**) Lipid accumulation was determined in en face aortas with Sudan-IV staining in high-fat diet fed baseline, LiαKO, LiβKO, and LiαβKO mice with and without tamoxifen treatment. Quantification of lipid content (lesional area) within (**B**) aortic arch, (**C**) thoracic aorta, and (**D**) total aorta. n = 7–9; mean ± SEM; * *p* < 0.05, ** *p* < 0.01, *** *p* < 0.001 (one-way ANOVA). Tam, tamoxifen; BL, baseline.

**Figure 3 ijms-23-09293-f003:**
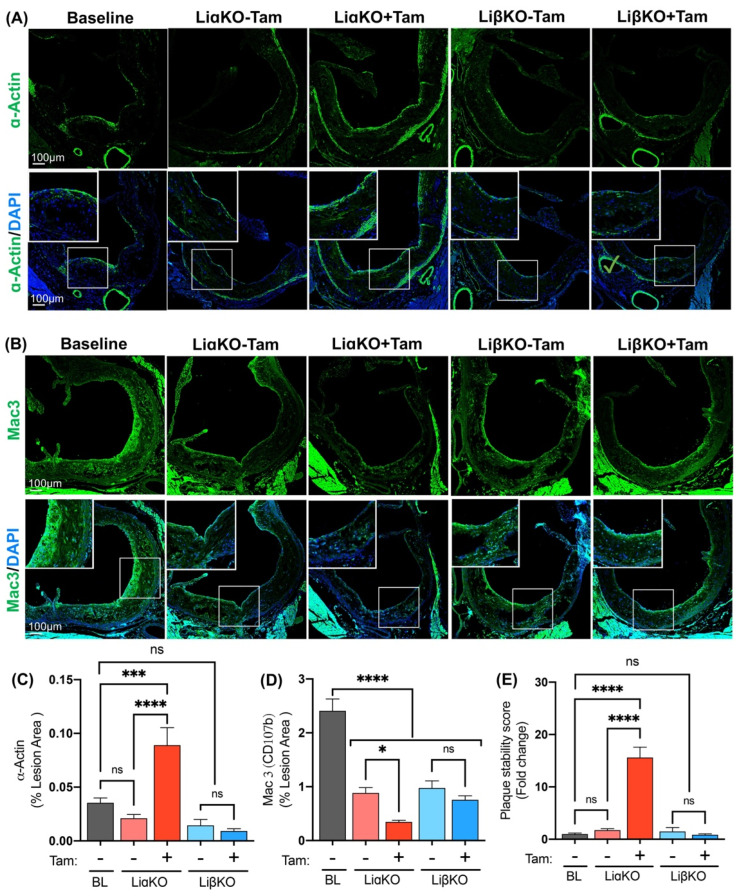
Macrophage-specific GSK3α-deficiency promotes a phenotype associated with atherosclerotic plaque stability in female Ldlr^−/−^ mice. Representative sections of aortic sinus immunostained with an antibody against (**A**) CD107b (Mac3, green) for macrophages and (**B**) α-actin (green) for smooth muscle cells. Quantification of (**C**) CD107b/Mac3 and (**D**) α-actin stained area normalized to the total lesion area. (**E**) Quantification of plaque stability score (SMC area/macrophage area). n = 9–13; mean ± SEM; * *p* < 0.05, *** *p* < 0.001, **** *p* < 0.0001 (one-way ANOVA). ns, not significant; Tam, tamoxifen; BL, baseline.

**Figure 4 ijms-23-09293-f004:**
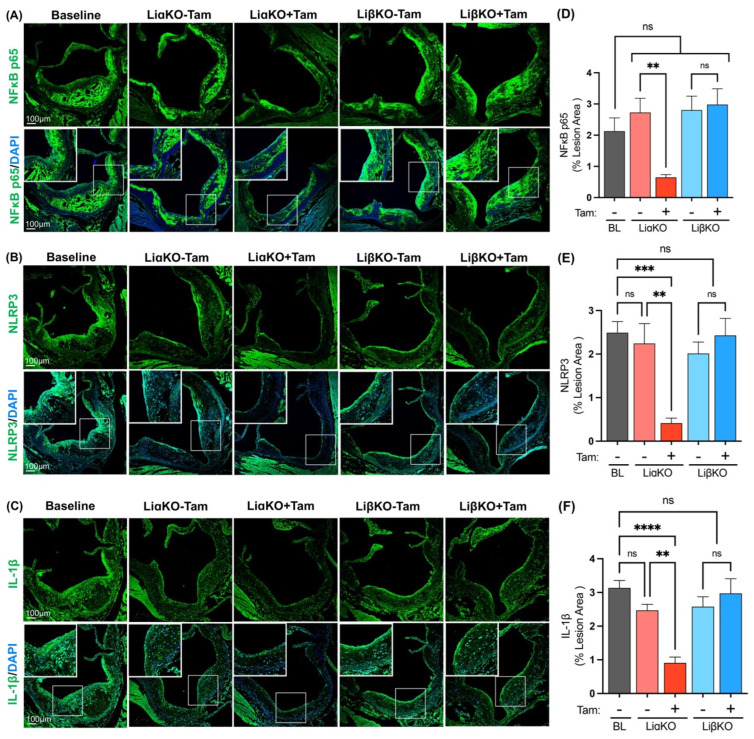
Reduced pro-inflammatory response in female *GSK3α*-deficient Ldlr^−/−^ mice. Representative sections of aortic sinus immunostained with an antibody against (**A**) NFκB (green), (**B**) NLRP3 (green), or (**C**) IL-1β (green). Quantification of (**D**) NFκB p65 and (**E**) NLRP3 and (**F**) IL-1β stained area normalized to the total lesion area. n = 9–13; mean ± SEM; ** *p* < 0.01, *** *p* < 0.001, **** *p* < 0.0001 (one-way ANOVA). ns, not significant; Tam, tamoxifen; BL, baseline; NFκB, nuclear factor kappa-light-chain-enhancer of activated B cells; NLRP3, NLR family pyrin domain containing 3; IL-1β, interleukin-1β.

**Figure 5 ijms-23-09293-f005:**
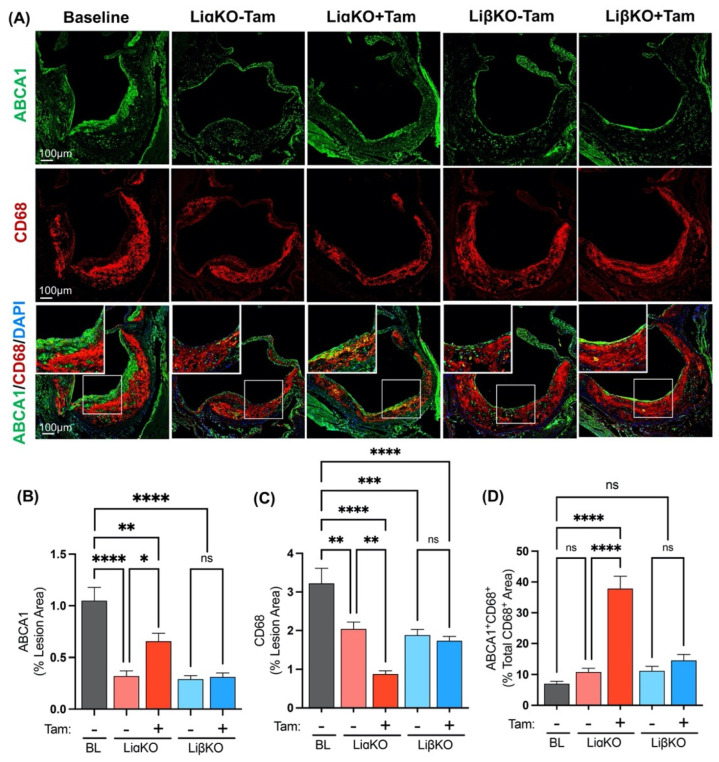
Elevated ABCA1 expression in GSK3α-deficient macrophages from female Ldlr^−/−^ mice. (**A**) Representative sections of aortic sinus immunostained with an antibody against ABCA1 (green) and co-stained with CD68 (red). Quantification of (**B**) ABCA1 and (**C**) CD68 area normalized to the total lesion area. (**D**) Quantification of co-stained (ABCA1 and CD68, yellow) area normalized to the total CD68 (macrophage, red) stained area. n = 5–7; mean ± SEM; * *p* < 0.05, ** *p* < 0.01, *** *p* < 0.001, **** *p* < 0.0001 (one-way ANOVA). ns, not significant; Tam, tamoxifen; BL, baseline; ABCA1, ATP-binding cassette transporter.

**Figure 6 ijms-23-09293-f006:**
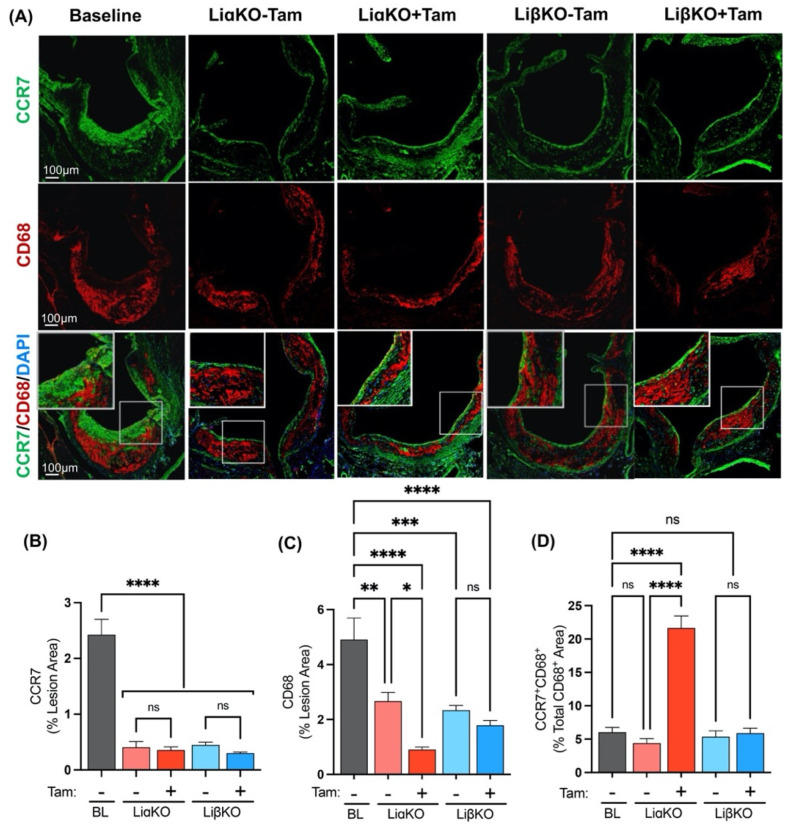
Female GSK3α-deficient *Ldlr*^−/−^ mice have elevated CCR7 expression. (**A**) Representative sections of aortic sinus immunostained with an antibody against CCR7 (green) and co-stained with CD68 (red). Quantification of (**B**) CCR7 and (**C**) CD68 area normalized to the total lesion area. (**D**) Quantification of co-stained (CCR7 and CD68, yellow) area normalized to the total CD68 (macrophage, red) stained area. n = 9–13; mean ± SEM; * *p* < 0.05, ** *p* < 0.01, *** *p* < 0.001, **** *p* < 0.0001 (one-way ANOVA). ns, not significant; Tam, tamoxifen; BL, baseline; CCR7, C-C chemokine receptor type 7.

## Data Availability

Not applicable.

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
