# Peer review of "Deletion of Macrophage-Specific Glycogen Synthase Kinase (GSK)-3α Promotes Atherosclerotic Regression in Ldlr−/− Mice"

_ijms, 2022, doi:10.3390/ijms23169293_

Round 1

Reviewer 1 Report

The authors have developed an in vivo model to study the effect of GSK3a/b on the regression of atherosclerotic plaques in Ldlr-KO mice using Cre-lox technology.

The study is very well conducted and results clearly presented.

Some minor aspects should be considered:

1.- In line 63, Reference #16 does not belong to the authors.

2.- Line 174: To clarify that are “female Ldlr-/- mice". In this way, in Discussion Section (Lines 314-316) is mentioned that: “Because of this observed sex-difference, the majority of the subsequent analysis was performed on the regressive plaques of female mice”; this phrase should be included in results to clarify that all experiments were performed on female mice.

3.- In Figure 2-B, y-axis legend, change “Lesin Area” by “Lesion Area”.

4.- The difference between α and β variants of GSK3 in this model need to be more deeply discussed. For example, the author to do mention that “GSK3α/β is a direct substrate of Akt and is inactivated by this kinase” but this association is shown form GSK3β more than GSK3α. It would be important to broaden the discussion on possible molecular mechanisms that mark functional differences between GSK3α and GSK3ß in relation to the atherosclerosis plaque regression results reported in this study.

Author Response

Response to Reviewers

The authors would like to thank the Editorial Board and the Reviewers for their constructive and helpful comments. We have addressed each comment as indicated in the point-by-point response presented below. For completeness, the reviews are posted in their entirety (in italics). Responses, where appropriate, are indicated by the symbol â–º.

1.- In line 63, Reference #16 does not belong to the authors.

â–ºIn order to clarify this issue the sentence in (line 62,63), has been changed to read,  

Our lab and others have previously shown that vascular endoplasmic reticulum (ER) stress activates glycogen synthase kinase (GSK3) to promote atherosclerotic development [16–18].”

2.- Line 174: To clarify that are “female Ldlr-/- mice". In this way, in Discussion Section (Lines 314-316) is mentioned that: “Because of this observed sex-difference, the majority of the subsequent analysis was performed on the regressive plaques of female mice”; this phrase should be included in results to clarify that all experiments were performed on female mice.

â–ºChange in the title as below (Line 177,178):

“2.4 Macrophage-specific GSK3α-deficiency promotes a phenotype associated with increased atherosclerotic plaque stability in female Ldlr-/- mice”

â–ºWe have added following sentence to result section (Line 165):

“Further characterization of atherosclerotic lesions was carried out in the female mouse model.”

3.- In Figure 2-B, y-axis legend, change “Lesin Area” by “Lesion Area”.

â–º This has been corrected in the new revised manuscript (Figure 2).

4.- The difference between α and β variants of GSK3 in this model need to be more deeply discussed. For example, the author to do mention that “GSK3α/β is a direct substrate of Akt and is inactivated by this kinase” but this association is shown form GSK3β more than GSK3α. It would be important to broaden the discussion on possible molecular mechanisms that mark functional differences between GSK3α and GSK3ß in relation to the atherosclerosis plaque regression results reported in this study.

â–ºWe have added following sentences to discussion (Line 358-364):

“Recent evidence suggests that GSK3α, but not GSK3β, modulates JAK-STAT pathway by indirectly affecting STAT phosphorylation [30]. Previously we have shown that GSK3α specifically regulates STAT3/6 phosphorylation/activation and NF-κB expression in macrophages. This suggests that GSK3α, but not GSK3β,actively plays a role in M1 macrophage polarization and pro-inflammatory response. The upstream mechanisms by which GSK3α and GSK3β may be differentially regulated are not fully understood.”

Reviewer 2 Report

The manuscript reports an interesting data. The claims are convincing and supported by the experimental data. The experimental procedures appear to be well done. The authors provided sufficient methodological detail that the experiments could be reproduced. The claims are appropriately discussed.  However, the authors should consider minor adjustments before publishing.   1.In the first part of Abstract authors say that male and female mice were studied (used in experiments), but the second part of Abstract  suggest that only female mice were studied. This is little confusing. Please edit slightly the Abstract.   2. Since that there are some differences in results obtained using female and male mice, it seems to me that legends to some Figures requires correction. Figures 1-3 clearly indicate that results were obtained using female mice. How about the results presented on Figures 4-5 ?   3. Authors wrote that (…) Tamoxifen  treatment did not significantly affect GSKα or GSKβ protein expression in other tissues, including muscle and liver (…) Page 2, lines 95 and 96. On page 3, lines 106, 107 authors wrote that (…) Tamoxifen – treated female mice had significantly lower total adipose tissue weight compared to controls (…). I wonder what is the effect of tamoxifen on GSKα or GSKβ in adipose tissue. Moreover, what does mean  (…) total adipose tissue weight (…) ? In Materials and Methods authors wrote (…) perigonadal fat pads were harvested an and weighed (…) Page 12, line 384.    

Author Response

Response to Reviewers

The authors would like to thank the Editorial Board and the Reviewers for their constructive and helpful comments. We have addressed each comment as indicated in the point-by-point response presented below. For completeness, the reviews are posted in their entirety (in italics). Responses, where appropriate, are indicated by the symbol â–º.

1.In the first part of Abstract authors say that male and female mice were studied (used in experiments), but the second part of Abstract suggest that only female mice were studied. This is little confusing. Please edit slightly the Abstract.  

â–º We have made the following changes to avoid this confusion (Line 13,14):

“Five-week-old low-density lipoprotein receptor knockout (Ldlr-/-) mice were fed high fat diet for 16 weeks to promote atherosclerotic lesion formation.” 

2.Since that there are some differences in results obtained using female and male mice, it seems to me that legends to some Figures requires correction. Figures 1-3 clearly indicate that results were obtained using female mice. How about the results presented on Figures 4-5 ?

â–º We have revised the figure legends for Figures 4-5 as described below:

Revised Figure 4: Reduced pro-Inflammatory response in female GSK3α-deficient Ldlr-/- mice.

Revised Figure 5. Elevated ABCA1 expression in GSK3α-deficient macrophages from female Ldlr-/- mice.

Revised Figure 6. Female GSK3α-deficient Ldlr-/- mice have elevated CCR7 expression.

3.Authors wrote that (…) Tamoxifen  treatment did not significantly affect GSKα or GSKβ protein expression in other tissues, including muscle and liver (…) Page 2, lines 95 and 96. On page 3, lines 106, 107 authors wrote that (…) Tamoxifen – treated female mice had significantly lower total adipose tissue weight compared to controls (…). I wonder what is the effect of tamoxifen on GSKα or GSKβ in adipose tissue. Moreover, what does mean  (…) total adipose tissue weight (…) ? In Materials and Methods authors wrote (…) perigonadal fat pads were harvested an and weighed (…) Page 12, line 384.

â–º Yes, we have shown that tamoxifen has no effect on GSKα or GSKβ protein expression in muscle and liver.We haven’t checked the effect of tamoxifen on GSKα or GSKβ expression in adipose tissue specifically. Tamoxifen has previously been shown to affect adipose tissue, independent of GSKα/β. The observed reduction in adipose weight is likely due to tamoxifen’s action as a selective estrogen receptor modulator [see Ref: Zhao, L., Wang, B., Gomez, N. A., de Avila, J. M., Zhu, M. J., & Du, M. (2020). Even a low dose of tamoxifen profoundly induces adipose tissue browning in female mice. International Journal of Obesity44(1), 226-234.]. This study suggests that tamoxifen increases thermogenesis in female mice, which leads to decrease in adipose weight.

â–ºin order to clarify what adipose tissue was being quantified, we have changed “total adipose” to “perigonadal adipose tissue” (Page 12, Line 407).